# The Cardioprotective Effect of *Magnolia officinalis* and Its Major Bioactive Chemical Constituents

**DOI:** 10.3390/ijms26094380

**Published:** 2025-05-05

**Authors:** Beata Olas

**Affiliations:** Department of General Biochemistry, Faculty of Biology and Environmental Protection, University of Lodz, Pomorska 141/143, 90-236 Lodz, Poland; beata.olas@biol.uni.lodz.pl; Tel./Fax: +48-42-635-4485

**Keywords:** *M. officinalis*, magnolol, honokiol

## Abstract

The genus *Magnolia* has been found to exhibit different biological properties, including antioxidant, anticancer, and others. For example, *Magnolia officinalis* is a classical traditional herb used in various Asian countries, especially China, South Korea, and Japan. Magnolia bark is the main medicinal part of this plant. This paper reviews the current state of knowledge regarding the *M. officinalis* bark and its active constituents, especially magnolol and honokiol, with a special emphasis on their cardioprotective activity in various models. This review paper also sheds new light on the cardioprotective mechanisms of magnolol and honokiol. However, their cardioprotective potential is limited to animal in vivo models and in vitro models. Only a single study has examined the cardiovascular properties of *M. officinalis* extract in obese mice. In addition, there is no clinical evidence for the absorption and bioavailability of *M. officinalis* extracts and their main bioactive compounds in humans. Moreover, there are no studies simultaneously comparing the activity of magnolol and honokiol. Therefore, there is a need for such studies. There are also no recommendations regarding their effective or safe doses for prophylaxis and the treatment of CVDs.

## 1. Introduction

The genus *Magnolia* has been found to exhibit different biological properties, including antioxidant, anticancer, and others [1,2]. For example, *Magnolia officinalis* is a classical traditional herb used in various Asian countries, especially China, South Korea, and Japan. Magnolia bark is the main medicinal part of this plant. For example, the *M. officinalis* cortex (the dried bark of *M. officinalis*, also known as “Houpo” (Chinese)) is widely used in traditional Chinese medicine. The magnolia flower is regarded as a symbol of purity and sincerity in China. In Korea, these flowers are afforded the status of a national symbol deployed for propaganda purposes [2,3,4,5,6,7,8,9,10]. More than 200 chemical compounds, including phenolic compounds, alkaloids, steroids, and volatile oils, have been isolated and identified in *M. officinalis* bark. Among these chemical compounds, the phenolic compounds magnolol (5,5″-diallyl-biphenyl-2,2″-diol) and honokiol (5,3′-diallyl-biphenyl-2,4′-diol) are the main well-characterized bioactive components isolated from *M. officinalis*, and they deserve special attention. They are stripped from the stems, branches, and roots of various genera of Magnolia—not only from *M. officinalis* but also *M. obovate* and *M. grandiflora.* They contain 1–10% of dry bark, depending on the Magnolia species and extraction method [3,4,5,6,7,8,9,10].

Honokiol is a phenylpropanoid compound, belonging to the class of neolignanes, characterized by the presence of a para-allyl-phenol and an ortho-allyl-phenol, joined together through ortho and para-C-C coupling. Structure–activity relationships have demonstrated the important roles of the 3′-allyl group 4′-hydroxyl and the 5-allyl groups for its biological properties, including cytotoxicity. Based on the basic chemical structure of honokiol, different derivatives with anti-cancer activity have been synthesized [9,11,12,13,14]. The concentration of honokiol in *M. officinalis* powder was found to be about 17–19 mg/g [15].

Magnolol is a polyphenolic binaphtalene compound and a structural isomer of honokiol. It is often consumed by humans in their daily diets, because it is the major component of extracts added to gums and mints [1,2,9,16,17].

Magnolol and honokiol have emerged as multifunctional compounds with many potential therapeutic properties, including neuroprotective, anticancer, antioxidant, and anti-inflammatory properties [2,18,19,20,21,22]. Moreover, these two chemical compounds are often used as markers to analyze the intracorporeal processes of single herbs or compound herbal formulae. They were also identified in plasma after rats consumed orally turbid liquor from *M. officinalis* [2].

In addition, honokiol and magnolol are known as cardioprotective substances, but only a few review papers have been published on this topic [6,16,23,24]. The chemical structure of the major bioactive constituents of *M. officinalis* bark is presented in Figure 1.

The aim of the present review is to provide an overview of the cardioprotective potential of extracts from *M. officinalis* bark and their chemical constituents—not only magnolol and honokiol but also their derivatives in various models. In addition, the cardioprotective mechanisms of *M. officinalis* and its main chemical constituents are reviewed in this paper to provide a basis for further study and development. It should be noted that cardiovascular diseases (CVDs) represent a large class of diseases, including hypertension, arrhythmia, heart disease, coronary artery disease, and others, and the incidence of CVDs is gradually increasing.

## 2. Research Methods

The published literature about *M. officinalis* and its main chemical compounds with cardioprotective potential was collected from various scientific databases, including ScienceDirect, Web of Science, PubMed, SCOPUS, Web of Knowledge, Elsevier, Google Scholar, and Sci Finder. The search terms comprised the terms “*M. officinalis*”, “*magnolia*”, “magnolol”, “honokiol”, and “cardioprotection” and their combinations. No time criteria were applied to the search, but recent papers were evaluated first. The last search was run on 1 April 2025. Papers were first selected based on their relevance to the title of the present manuscript, and the identified articles were screened by reading the abstracts. Any relevant identified articles were summarized. About 203 articles were obtained from the searches, and only 97 were included in this review. The cardioprotective potential of *M. officinalis* and its major bioactive chemical constituents (especially magnolol and honokiol) in various models is summarized, and the current studies are discussed.

## 3. Cardiovascular Properties of *M. officinalis* Extract and Its Bioactive Compounds

### 3.1. Cardioprotective Actions (In Vitro and In Vivo Models)

The review paper of Ho and Hong [16] demonstrated that magnolol changes the functions of various cells in the cardiovascular system. These effects are dose-dependent and are the consequences of different molecular mechanisms induced by magnolol. For example, the results of Hong et al. [25] indicate that magnolol (10^−5^ mg/kg) significantly reduced the infarct size. The intravenous injection of magnolol (>10^−6^ mg/kg) before coronary artery ligation inhibited both reperfusion- and ischemia-induced ventricular tachycardia and ventricular fibrillation. Other results demonstrated that magnolol (10^−4^ to 10^−3^ mg/kg) restored systolic wall thickening [26].

Another study showed that magnolol reduced pulmonary arterial hypertension by inhibiting the expression of endothelin-1 and angiotensin II and decreased sight ventricular hypertrophy and pulmonary vascular remodeling induced by hemodynamic changes in rats. In addition, this compound restored the expression of Sirt3, β-catenin, and vascular endothelial cadherin (VE cadherin) [27]. Liang et al. [28] observed that magnolol (100 mg/kg/day, for 3 weeks) modulated insulin-induced aortic vasodilation to delay the development of hypertension by restoring insulin-induced protein kinase B (also known as AKT) and endothelial nitric oxide synthase (eNOS) activation, decreasing tribbles 3, and increasing PPARγ in young spontaneously hypertensive rats. In vivo, magnolol (100 mg/kg, i.p. injection) downregulated the expression of endothelin-1, inducible nitric oxide synthase (iNOS), and O_2_^−^ production and upregulated angiotensin II [29].

In cultured human umbilical vein endothelial cells (HUVECs), magnolol incubation increased PPARγ, decreased tribbles homolog 3 (TRB3) expression, and restored insulin-induced phosphorylated Akt and eNOS levels and nitric oxide (NO) production, which was blocked by both PPARγ antagonists and siRNA targeting PPARγ. The improved insulin signaling in HUVECs caused by magnolol was abolished by upregulating TRB3 expression. In another in vitro model, magnolol inhibited the phosphorylation of JNK/p38, the translocation of human antigen R (HuR), the activation of nuclear factor kappa-light-chain-enhancer of activated B cells (NF-κB), and the expression of vascular cell adhesion molecules 1 (VCAM-1), leading to a reduction in leukocyte adhesion in tumor necrosis factor-α (TNF-α)-treated human aortic endothelial cells [30].

Pillai et al. [31] also noted the antihypertrophic action of honokiol, which increased the expression of the deacetylase SIRT3 in myocardial mitochondria. In addition, honokiol (0.2 mg/kg/day, for 5 days) inhibited oxidative stress and cardiac inflammation and suppressed cardiotoxicity stimulated by doxorubicin (5 mg/kg/week) in mice [32]. This compound could also upregulate the expression of PPARγ in the mouse heart [32]. According to Zhang et al. [33], 4-*O*-methylhonokiol (0.5 and 1 mg/kg b.w.) exhibited cardioprotective potential, associated with the activation of the nuclear factor erythroid-2 (Nrf-2) and protein kinase B (PKB/AKT) signaling pathways and the inhibition of cardiac lipid accumulation mediated by CD36, in a high-fat-diet mouse model. Lipid peroxidation in heart tissue was measured by the thiobarbituric acid reactive substances (TBARS) assay, and the levels of phosphorylated p-Nrf2 were measured by Western blotting. The cardioprotective effects of various bioactive constituents of *M. officinalis* were also described by other authors, although the mechanism is still not clear [1,33,34].

Moreover, Sun et al. [35] observed that *M. officinalis* extract (containing 12% magnolol, 14.2% honokiol, and other compounds) improved insulin resistance caused by a high-fat diet through decreases in cardiac hypertrophy and dysfunction, cardiac oxidative stress, and inflammation or cardiac lipid accumulation. In this model, mice were fed a high-fat (60 kcal% fat) diet for 24 weeks to induce obesity. These mice were also given a vehicle, 2.5, 5, or 10 mg/kg b.w. *M. officinalis* extract via gavage daily. The three doses of the used extract slightly ameliorated insulin resistance without a decrease in body weight gain induced by high-fat diet feeding. The used extract at 10 mg/kg slightly attenuated the mild cardiac hypertrophy and dysfunction induced by high-fat diet feeding. Both 5 mg/kg and 10 mg/kg of plant extract treatment significantly inhibited cardiac lipid accumulation. In addition, the used extract had anti-apoptotic action via the mitochondria pathway and AF/caspase-8/Bax/Bcl-2 biochemical cascade. In this experiment, the *M. officinalis* extract was prepared by Bioland Co., Ltd. (Korea) and dissolved in 0.5% ethanol.

### 3.2. Anti-Obesity Actions (In Vitro and In Vivo Models)

Obesity is an important cause of various cardiovascular diseases, bringing about hypertension, cardiac morphological changes, and insulin resistance; it also increases oxidative stress and inflammation in the heart. The control of obesity is an important target for the prevention or treatment of CVDs. In addition, various plant preparations, including extracts, have anti-obesity potential [36,37].

Yimam et al. [38] indicated that an extract from *Morus alba, Yerba mate*, and *M. officinalis* had a significant effect on weight loss in high-fat mice. In this experiment, the plant extract was administered at oral doses of 300 mg/kg, 450 mg/kg, and 600 mg/kg for 7 weeks; orlistat (40 mg/kg/day) was used as a positive control. Statistically significant changes in body weight (decreased by 9.1, 19.6, and 25.6% compared to the control group at week 7) were observed for mice treated with the used extract at 300, 450, and 600 mg/kg, respectively. Decreases in total cholesterol, triglycerides, and low-density lipoprotein (LDL) were observed for the plant extracts at all three doses (300, 450, and 600 mg/kg). In this experiment, *M. officinalis* stem bark was extracted via a supercritical fluid and further crystalized to give a mixture of magnolol and honokiol with content higher than 95%. However, this study, where three extracts were administered simultaneously, did not indicate the activity of any of them separately. The activity may have been demonstrated by each of them individually, or it may have been the effect of interactions between the components (positive or negative); therefore, it is difficult to indicate their anti-obesity activity of the *M. officinalis* extract.

A diet with 0.02% honokiol and magnolol (for 16 weeks) did not change the body weights of high-fat-diet-fed obese mice, but this diet significantly decreased the white adipose tissue weight and fat cell size. These effects were associated with increases in energy expenditure and adipose fatty acid oxidation and decreases in fatty acid synthase activity and the expression of genes related to fatty acid synthesis, desaturation, and uptake, as well as adipocyte differentiation in white adipose tissue. Moreover, honokiol improved the plasma adiponectin concentration [39]. Similar results were noted for 4-*O*-methylhonokiol, which reduced significantly the levels of lipids such as plasma triglyceride and cholesterol. This compound also protected against high-fat-diet-induced obesity and systemic insulin resistance in mice [39]. In addition, Kim and Jung [40] observed that honokiol inhibited lipogenic enzymes in adipose tissue in type 2 diabetic mice, and it led to significant decreases in the adipose tissue weight. In this study, male C57BL/KsJ-db/db mice were fed a normal diet with or without honokiol (0.02%, w/w) for 5 weeks. Other experiments demonstrate that honokiol activates the liver kinase B1-AMP-activating protein kinase (LKB1-AMPK) signaling pathway and attenuates lipid accumulation in hepatocytes. In their study, Seo et al. [41] elucidated the cellular mechanism by which honokiol alleviates the development of non-alcoholic steatosis. HepG2 cells were treated with honokiol (for 1 h) and then exposed to 1 mM free fatty acid (FFA, for 24 h) to simulate non-alcoholic steatosis in vitro. In addition, C57BL/6 mice were fed with a high-fat diet for 28 days, and honokiol (10 mg/kg/day) was administered daily. Honokiol concentration-dependently attenuated intracellular fat overloading and triglyceride accumulation in free fatty acid-exposed HepG2 cells. These effects were blocked by pretreatment with an AMPK inhibitor. Honokiol significantly inhibited sterol-regulatory element-binding protein-1c (SREBP-1c) maturation and the induction of lipogenic proteins, stearoyl-CoA desaturase-1, and fatty acid synthase in FFA-exposed HepG2 cells. Honokiol also induced AMPK phosphorylation and subsequent acetyl-CoA carboxylase phosphorylation, which were inhibited by the genetic deletion of LKB1. Moreover, honokiol attenuated the increases in the hepatic triglyceride and lipogenic protein levels and fat accumulation in the mice fed with a high-fat diet, while significantly inducing LKB1 and AMPK phosphorylation.

Parray et al. [42] noted that magnolol decreased reactive oxygen species (ROS) generation; upregulated the expression of SIRT1, UCP1, Tbxl, and others; and downregulated the expression of SREBP1 and FAS, resulting in the browning of 3T3-L1 adipocytes, enhancing lipolysis, and repressing oxidative stress through the AMPK, protein kinase A (PKA), and PPARγ signaling pathways. Recently, Chu et al. [43] also noted that magnolol and honokiol promote adipose tissue browning and resist obesity through modulating peroxisome proliferator-activated receptor alpha/gamma (PPARα/γ) activity. Additionally, magnolol (1–100 µM) blocked homocysteine-induced endothelial dysfunction in porcine coronary arteries (separated from the hearts of pigs) [44].

### 3.3. Antiatherosclerotic Actions (In Vitro and In Vivo Models)

Atherosclerosis is considered the most common potential pathological process in various CVDs, and it is associated with the inflammatory process, which is characterized by the overexpression of cyclooxygenase-2 and inducible nitric oxide synthase and the excessive synthesis of prostaglandins and nitric oxide. Nuclear factor-κB (NF-κB) and mitogen-activated protein kinase (MAPK) are the most crucial signaling pathways in the inflammatory process. Different authors indicate that magnolol and honokiol have antiatherosclerotic properties by inhibiting the inflammatory process [45,46,47,48]. For example, Ou et al. [45] observed that honokiol (2.5–100 µM) reduced copper-induced LDL oxidation modification to reduce ox-LDL production, as demonstrated by diene formation and the TBARS assay. Honokiol also regulated eNOS protein expression and monocyte macrophage adhesion to regulate ox-LDL-induced vascular endothelial dysfunction. The expression of adhesion molecules, including ICAM, VCAM, and E-selectin, and eNOS affected by oxLDL was investigated by Western blotting and flow cytometry. According to the research of Karki et al. [47], vascular cell adhesion molecule-1 migration was inhibited by magnolol through the cytoskeletal remodeling pathway. Another means by which to mitigate the inflammatory process is inhibiting ROS production, ERK1/2 phosphorylation, and NF-κB activation via magnolol (10–40 µM). Magnolol also inhibited the mRNA and protein expression of cyclins D1 and E, CDK4, and CDK2 in vivo and in vitro. Moreover, the oral administration of magnolol (100 mg/kg/day) significantly inhibited the intimal area and intimal/medial ratio. Data from Wu et al. [48] demonstrated also that magnolol (5–20 µM) inhibited the proliferation of vascular smooth muscle cells and DNA synthesis in rats, without causing cell cytotoxicity. A flow-cytometric analysis demonstrated that magnolol inhibited the S-phase entry of vascular smooth muscle cells and stimulated platelet-derived growth factor (PDGF). The authors also noted that magnolol caused this effect by inhibiting the mRNA and protein expression of cyclins D1 and E and cyclin-dependent kinases 2 and 4. Other papers also indicate that *M. officinalis* extract and its selected chemical compounds, including magnolol, honokiol, and their derivatives, has anti-inflammatory effects in various in vitro and in vivo models [49,50,51,52].

Besides the inflammatory process, oxidative stress also plays an important role in the progression of CVDs, including atherosclerosis. A few papers describe the antioxidant properties of the components of *M. officinalis*, especially magnolol and honokiol [53,54,55,56,57]. Firstly, Lo et al. [58] noted the cardioprotective potential of magnolol through its antioxidant properties. Chen et al. [59] observed that the inhibition of lipid peroxidation by magnolol (10–3 mg/kg) played an important role in preventing vessel restenosis in cholesterol-fed rabbits. In addition, not only magnolol but also honokiol was effective in reducing peroxynitrite (ONOO^−^) [60], which acts as a reactive nitrating and nitrosating agent and a gasomediator in hemostasis and thrombosis [61]. Amorati et al. [56] found that the antioxidant activity of these compounds is independent of the presence of allyl groups, and intermolecular and intramolecular interactions influence their antioxidant potential.

### 3.4. Anti-Platelet Actions (In Vitro and In Vivo Models)

*M. officinalis* extracts and their components may also modulate hemostasis and have an inhibitory effect on blood platelet activation. Honokiol and magnolol obtained from *M. officinalis* showed anti-platelet potential, including anti-aggregatory properties [22,62,63]. Honokiol and magnolol inhibited blood platelet aggregation and ATP release in rabbit platelet-rich plasma induced by physiological agonists (arachidonic acid or collagen), without affecting that induced by thrombin, ADP, or platelet-activating factor (PAF). The aggregation of washed blood platelets was more markedly inhibited than that of platelet-rich plasma, while the aggregation of whole blood was least affected by both inhibitors. Thromboxane B_2_ formation caused by collagen thrombin or arachidonic acid was, in each case, inhibited by honokiol and magnolol.

Other studies indicate that magnolol (20–60 μM) dose-dependently also inhibited intracellular calcium mobilization in blood platelets. Its anti-platelet action was blocked by inhibitors of the peroxisome proliferator-activated receptor: a selective PPAR-β antagonist (GSK0660) or a PPAR-γ antagonist (GW9662) [53]. By upregulating PPARβ/γ activity and expression, magnolol activated the AKT/eNOS/NO/cGMP/PKG cascade reaction and inhibited cyclooxygenase-1 (COX-1), intracellular calcium mobilization, and the protein kinase C (PKC) pathway. Additionally, collagen- or arachidonic acid-stimulated thromboxane B_2_ formation and elevations in COX-1 activity caused by arachidonic acid were also markedly attenuated by magnolol. Similarly, the administration of PPAR-β/γ antagonists remarkably abolished the actions of magnolol in preventing platelet plug formation and prolonging the bleeding time in mice [53].

The anti-platelet properties have been determined using various methods; for example, blood platelet aggregation was measured via turbidimetry in platelet-rich plasma or washed blood platelets [53,62,63]. The key functional ingredients in *M. officinalis* bark with known cardioprotective activity and the potential molecular mechanisms behind its cardioprotective actions are given in Figure 1. This figure shows that, for example, magnolol and honokiol demonstrate anti-inflammatory and anti-platelet effects on various models by regulating signaling pathways, such as COX-1, PKC, AMP-activating protein kinase, and others.

The cardioprotective actions of *M. officinalis* extracts and their main bioactive compounds—magnolol, honokiol, and 4-methoxyhonokiol—in various models are described in Table 1. Table 1 demonstrates that the cardioprotective potential is limited to animal in vivo models and in vitro models. In addition, only a single study has examined these properties of *M. officinalis* extracts in obese mice.

## 4. The Bioavailability and Safety of *M. officinalis* Extract and Its Bioactive Compounds

According to the Chinese Pharmacopoeia (version 2020), the recommended dosage of *M. officinalis* is 3–10 g [74]. In addition, the safety of *M. officinalis* and its main bioactive compounds, especially honokiol and magnolol, has been discussed by various authors [2,5,75]. For example, Liu et al. [75] noted that there were no adverse effects of *M. officinalis* for dosages above 240 mg/kg/day (for 90 days) in rats. Moreover, increasing doses of 625–2500 mg/kg/day of *M. officinalis* extract (containing 1.5% honokiol and 94% magnolol, for 14 days) were also safe in mice. A similar effect was observed by Li et al. [76]. The oral administration of capsuled extracts of *M. officinalis* and *Phellodendron amurense* (250 mg, three times/day, for 6 weeks) was well tolerated in both healthy and obese patients [77]. However, the oral administration of 5–10 g/kg of *M. officinalis* extract (for 14 days) decreased renal and liver function in rats, but the authors did not describe the phytochemical characteristics of the used extract and used solvent [78].

The safe dose of magnolol from adolescence is up to 1.64 mg/kg per day. However, due to its low water solubility and quick metabolism, it has low bioavailability [17]. Sheng et al. [79] indicate that the bioavailability of magnolol reaches about 17.5%, and the value of the mean peak plasma concentration of magnolol reaches about 426.4 ng/mL after oral administration. The liver contained the highest concentrations of magnolol and magnolol-glucuronides [80]. Importantly, various studies have demonstrated that magnolol has cytotoxicity [46,47,64,81,82,83,84,85]. For example, magnolol (40 µM) possessed cytotoxicity towards vascular smooth muscle cells [46,47].

The pharmacokinetics of honokiol are not yet defined in humans but are available in mice. For example, the injection of 5–10 mg/kg of honokiol exhibited a plasma t_1/2_ value of about 40–60 min [86]. Moreover, Bohmdorfer et al. [87] identified oxidation as the main phase I metabolic pathway of honokiol, with amino acid conjugation, acetylation, sulfation, and glucuronidation being the main phase II metabolic pathways, in rats. They also detected 51 metabolites: 5 for phase I and 32 for phase II.

Various formulations, such as nanoparticles, solid dispersions, emulsions, and liposomes, may ameliorate the water solubility and bioavailability of magnolol and honokiol [9,14,79,88,89,90,91,92,93,94,95]. Recently, Sampieri-Moran et al. [94] demonstrated that the delivery of *M. officinalis* bark extract in nanoemulsions formed by high- and low-energy methods improved the bioavailability of magnolol (increased by 3.03 times) and honokiol (increased by 3.47 times).

The potential health benefits of *M. officinalis* have led to the development of various products containing magnolia bark and its derivatives. Recently, Siudem et al. [96] studied the quality and quantity of seven dietary supplements (E1–E7) containing magnolia bark and its preparations, especially those available in the Polish market. These supplements had various forms: tablets, capsules, powder, and bark. The authors identified the active substances, magnolol and honokiol, in all tested preparations using high-performance liquid chromatography (HPLC) and nuclear magnetic resonance spectroscopy (^1^H NMR). In addition, their results indicate that both techniques provide similar results and can be used for the quality control of magnolia dietary supplements. For example, one tested supplement (E2, as capsules) contained 225 mg of *M. officinalis* bark extract (202.5 mg honokiol). Another supplement (E3, as capsules) contained 400 mg *M. officinalis*. A tested supplement (E5) as a powder had 40% honokiol and 50% magnolol [96].

## 5. Conclusions

The present work reviews the newly available literature regarding the cardiovascular potential of two individual compounds (magnolol and honokiol) isolated from *M. officinalis* bark, but this knowledge is limited to animal in vivo models and in vitro models. Only a single study has examined the cardiovascular properties of *M. officinalis* extract in obese mice. In addition, there is no clinical evidence for the absorption and bioavailability of *M. officinalis* extracts and their main bioactive compounds in humans. Moreover, there are no studies simultaneously comparing the activity of magnolol and honokiol. Generally, various studies have taken mixtures of these two phenolic compounds or the total phenolic compounds of *M. officinalis* as research objects. However, there are significant differences in the biological activity of these phenolic compounds. For example, Kuribara et al. [97] suggest that honokiol has stronger anti-anxiety effects than magnolol. Therefore, there is a need for such studies comparing the cardioprotective actions of magnolol and honokiol. There are also no recommendations regarding their effective or safe doses for prophylaxis and the treatment of CVDs.

Moreover, there are differences in the extraction processes and the relative amounts of the chemical compounds in *M. officinalis* extracts; therefore, it is very difficult to compare the dose–effect relationships to the biological materials, and this is a significant topic for further studies. Beyond the extraction methods, other factors like the growth environment, harvesting, and storage of the herb may influence its efficacy, thereby indirectly affecting the dosage.

No concrete clinical experiments have investigated the interactions of *M. officinalis* preparations, including supplements and their bioactive components, with various drugs used in the treatment of CVDs. In the future, a greater emphasis should be placed on human studies and clinical trials.

Despite this, this review paper sheds new light on the cardioprotective mechanisms of magnolol, honokiol, and their derivatives (for example, 4-*O*-methylhonokiol), but further studies are needed to clarify their mechanisms of action.

## Figures and Tables

**Figure 1 ijms-26-04380-f001:**
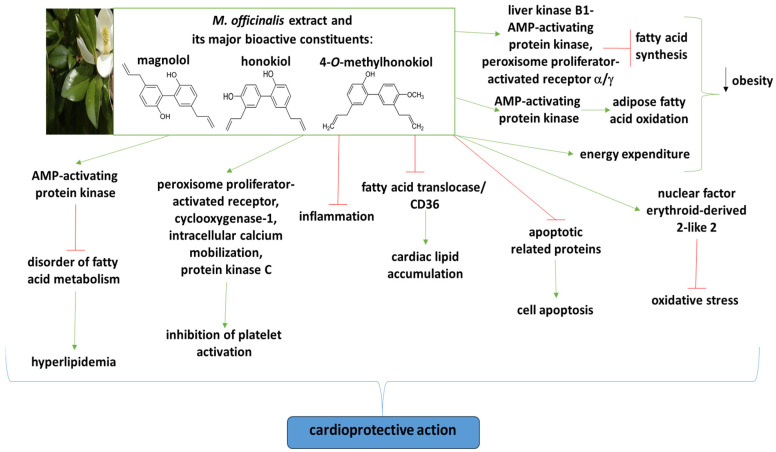
The chemical structure of the major bioactive constituents of *M. officinalis* bark and their potential cardioprotective mechanisms of action. More details regarding the roles of various key factors in the cardioprotective mechanism are described in the text.

**Table 1 ijms-26-04380-t001:** Cardioprotective action of *M. officinalis* extracts and their main bioactive compounds in various models.

Cardioprotective Action				
Extract/Bioactive Compound	Model	Dosage	Result	Reference
In vitro models				
Magnolol	Right coronary arteries from pig hearts	1–100 µM	Relaxed the coronary artery and inhibited COX-2 and iNOS protein expression	[44]
Magnolol	Human aortic smooth muscle cells	10–20 µM	Inhibited migration of tested cells by suppressing cytoskeletal remodeling and neointima formation	[47]
Magnolol	RAW 264.7 cells	25–100 µM	Reduced the expression of TLR2, the production of ROS, and inflammatory cytokines	[64]
Magnolol	Human aortic endothelial cells	5 µM	Reduced leukocyte adhesion	[30]
Magnolol	Vascular smooth muscle cells	5–20 µM	Suppressed proliferation of tested cells and DNA synthesis by inhibiting ROS generation, as well as the expression of cyclin D1/E and cyclin-dependent kinases 2 and 4	[48]
Magnolol	LPS-induced RAW 264.7 cells	15–60 µg/mL	Downregulating TLR4 expression, NF-κB and MAPK pathway activation, and proinflammatory cytokine excretion	[65]
Magnolol	Vascular smooth muscle cells	1–10 µM	Inhibited the expression of endothelin-1	[29]
Magnolol	Rabbit blood platelets	20–60 µM	Inhibited blood platelet activation and aggregation	[53]
Magnolol	Rat thoracic aorta	10–100 µg/mL	Inhibited vascular contractions	[63]
Magnolol	Endothelial cells	1–30 µM	Inhibited IL-6-induced STAT3 activation and gene expression	[49]
Magnolol	Isolated rat heart mitochondria	IC_50_ = 0.08 µM	Inhibited lipid peroxidation	[58]
Magnolol	Mouse 3T3-L1 preadipocytes	10–60 µM	Reduced triglyceride levels	[29]
Magnolol and honokiol	Rabbit blood platelets	1–100 µg/mL	Inhibited blood platelet activation, including arachidonic acid metabolism and intracellular calcium increase	[62]
Honokiol	Cardiomyocytes	5 and 10 µM	Activating mitochondrial Sirt3	[31]
Honokiol	H9c2 rat cardiomyocytes	2.5 and 5 µM	Activated PPARγ and suppressed mitochondrial protein deacetylation	[32]
Honokiol	Human umbilical vein endothelial cells	2.5–20 µM	Resisted oxLDL-induced cytotoxicity and adhesion molecule expression	[45]
Honokiol	Isolated rat heart mitochondria	IC_50_ = 0.1 µM	Inhibited lipid peroxidation	[58]
4-Methoxyhonokiol	RAW 264.7 cells	1–30 µM	Suppressed COX-2 and iNOS	[66]
In vivo models				
Extract (Bioland Co., Ltd. (Korea))	Obese mice	2.5–10 mg/kg	Reduced cardiac lipid accumulation	[35]
Magnolol	Obese mice	100 mg/kg	Reduced dyslipidemia	[67]
Magnolol	Human APOA5 knock-in mice	30 mg/kg	Reduced triglyceride levels	[29]
Magnolol	Mice with hyperlipidemia	10 and 20 mg/kg	Reduced dyslipidemia	[68]
Magnolol	Male Sprague Dawley rats	1–100 µg/kg	Reduced ventricular fibrillation and animal mortality (for 10 µg/kg) and reduced proportion of myocardial ischemic necrosis area	[47]
Magnolol	Mice with one first-order vein ligated	20 µg/kg	Inhibited venous remodeling	[69]
Magnolol	Rats with hypertension	100 mg/kg	Decreased blood pressure	[28]
Magnolol	Male Sprague Dawley rats	50 and 100 µg/kg	Increased luminal area and attenuated neointima formation, intimal area, and intimal/medial ratio	[46]
Magnolol	Male Sprague Dawley rats	10 mg/kg	Regulated angiotensin-converting enzyme/angiotensin II/Ang II type 1 receptor cascade and angiotensin-converting enzyme 2	[29]
Magnolol	Male spontaneous hypertensive rats	100 mg/kg	Decreased blood pressure through upregulated eNOS, Akt, and PPAR-γ and improved vascular insulin resistance	[30]
Magnolol	C57BL/6J mice	10–50 mg/kg	Decreased the expression of inflammatory cytokines	[70]
Magnolol	LPS-induced Sprague Dawley rats	10 and 20 mg/kg	Increased the expression of PPAR-γ, altered iNOS and COX-2 expression, ROS production, and proinflammatory factor concentrations	[71]
Magnolol	Coronary occlusion and reperfusion model in Sprague Dawley rats	0.001–0.1 µg/mL	Reduced infarct size and suppressed ventricular arrythmia	[25]
Magnolol	Rabbits with coronary artery occlusion	0.1 and 1 µg/kg	Protected myocardium against stunning	[26]
Magnolol	Rats with pulmonary arterial hypertension	10 mg/kg	Inhibited expression of angiotensin II	[29]
Magnolol	Cholesterol-fed rabbits	1 µg/kg	Suppressed monocyte chemoattractant protein-1 and intimal hyperplasia	[59]
Magnolol	Cholesterol-fed rabbits	1 µg/kg	Inhibited TNFα-induced intracellular adhesion molecule 1 expression	[72]
Honokiol	Mice with obesity	100 mg/kg	Reduced dyslipidemia	[67]
Honokiol	Hypertrophic mice	0.2 mg/kg	Reverse cardiac hypertrophy	[31]
Honokiol	Mice with cardiac injury induced by doxorubicin	0.2 mg/kg	Activated PPARγ and suppressed mitochondrial protein deacetylation	[32]
Honokiol	Mice with obesity	0.02% in diet	Reduced obesity	[73]
4-Methoxyhonokiol	Male ICR mice	20 and 100 mg/kg	Suppressed COX-2 and iNOS	[66]
4-Methoxyhonokiol	Mice with obesity	0.5 and 1 mg/kg	Reduced body fat and regulated lipid metabolism	[39]

## Data Availability

Not applicable.

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
