# Peer review of "The Cardioprotective Effect of Magnolia officinalis and Its Major Bioactive Chemical Constituents"

_ijms, 2025, doi:10.3390/ijms26094380_

Round 1
Reviewer 1 Report
Comments and Suggestions for Authors
This article mainly introduces the traditional uses, main chemical components (such as honokiol), structural characteristics and pharmacological activities of Magnolia officinalis. The effects of Magnolia officinalis extract and its active components on anti obesity, anti atherosclerosis and anti platelet aggregation were summarized in detail, and its mechanism (such as regulating AMPK, PPAR γ and other signaling pathways) was discussed. The pharmacokinetics and potential toxicity of Magnolia officinalis and its active ingredients were studied. The article cites a large number of literature research, with detailed data, strong argument, and clear structure, which is gradually expanded from foundation to application. In terms of innovation, this paper summarized the research progress of Magnolia officinalis in cardiovascular protection, discussed the bioavailability, safety and other issues, pointed out the problems of the current research, and provided some ideas for the follow-up research and development.
Suggestions: 1. supplement the discussion on the current situation of clinical research or put forward the direction of future clinical research
2. the abbreviations of some references are not standardized ("et" not italicized) 3. the list of abbreviations (COX-1) indicates the full name when it first appears. 4. the spacing of some paragraphs is different, and the adjustment is unified
Author Response
This article mainly introduces the traditional uses, main chemical components (such as honokiol), structural characteristics and pharmacological activities of Magnolia officinalis. The effects of Magnolia officinalis extract and its active components on anti obesity, anti atherosclerosis and anti platelet aggregation were summarized in detail, and its mechanism (such as regulating AMPK, PPAR γ and other signaling pathways) was discussed. The pharmacokinetics and potential toxicity of Magnolia officinalis and its active ingredients were studied. The article cites a large number of literature research, with detailed data, strong argument, and clear structure, which is gradually expanded from foundation to application. In terms of innovation, this paper summarized the research progress of Magnolia officinalis in cardiovascular protection, discussed the bioavailability, safety and other issues, pointed out the problems of the current research, and provided some ideas for the follow-up research and development.
Response: Thank you for reviewing the manuscript and providing such helpful comments. All of them have been taken into consideration when revising the manuscript.
Suggestions:
- supplement the discussion on the current situation of clinical research or put forward the direction of future clinical research
Response: I have added more information about it. For example, “This present work reviews newly-available literature regarding the cardiovascular potential especially two individual compounds (magnolol and honokiol) isolated from M. officinalis bark, but this potential is limited to animal in vivo models and in vitro models. Only a single study has examined the cardiovascular properties of M. officinalis extract in obese mice. In addition, there is no clinical evidence for the absorption and bioavailability of M. officinalis extracts and their main bioactive compounds in human. Moreover, there were not studies simultaneously comparing the activity of magnolol and honokiol. Generally, various studies have taken mixtures of these two phenolic compounds or total phenolic compounds of M. officinalis as research objects. However, there are significant differences in biological activity for these phenolic compounds. For example, Kuribara et al. [1998] suggest that honokiol has stronger anti-anxiety effects than magnolol. Therefore, there is a need for such studies comparing the cardioprotective action of magnolol and honokiol. There is also no recommendations regarding their effective or safe doses for the prophylaxis and treatment of CVDs.”
“No concrete clinical experiments have investigated the interaction of M. officinalis preparations, including supplements and their bioactive components with various drugs using in the prophylaxis and treatment of CVDs. In the future, more emphasis should be put on human studies and clinical trials.
- the abbreviations of some references are not standardized ("et" not italicized)
Response: I have not corrected, because according to instruction for authors. “et al.” should be nor italicized.
- the list of abbreviations (COX-1) indicates the full name when it first appears.
Response: I have corrected it.
- the spacing of some paragraphs is different, and the adjustment is unified
Response: I have corrected it.
Reviewer 2 Report
Comments and Suggestions for Authors
This review comprehensively summarizes the cardioprotective effects of Magnolia officinalis and its primary active constituents, magnolol and honokiol. Beginning with the traditional applications of M. officinalis, the article focuses on elucidating underlying mechanisms—including antioxidant, anti-inflammatory, anti-atherosclerotic, and antiplatelet activities—supported by in vitro experiments and animal models to validate its cardiovascular protective potential. The discussion is well-structured and logically developed, this manuscript is recommended for publication after minor revisons.
- Multiple spelling/grammatical errors exist (e.g., Line 61: "The are also identified…"; Line 321: "There is also nor…").
- Inconsistent italicization of M. officinalis; some instances are italicized while others are not.
- Formatting errors: Line 221 uses incorrect font ("another papers also indicate that…"); Line 187 misspells "4-O-methylhonokiol" as "4-0-methylhonokiol."
- Line 322: Beyond extraction methods, factors like growth environment, harvesting, and storage of the herb may influence efficacy, thereby indirectly affecting dosage.
- The figure oversimplifies mechanisms. Key factors (e.g., AMPK, NF-κB) and their directional roles in cardioprotection are not clearly labeled, despite being discussed in the text.
- The segment on "The bioavailability and safety of M. officinalis extract…" lacks explicit linkage to cardioprotection. Authors should clarify its relevance to the main theme.
Author Response
This review comprehensively summarizes the cardioprotective effects of Magnolia officinalis and its primary active constituents, magnolol and honokiol. Beginning with the traditional applications of M. officinalis, the article focuses on elucidating underlying mechanisms—including antioxidant, anti-inflammatory, anti-atherosclerotic, and antiplatelet activities—supported by in vitro experiments and animal models to validate its cardiovascular protective potential. The discussion is well-structured and logically developed, this manuscript is recommended for publication after minor revisons.
Response: Thank you for reviewing the manuscript and providing such helpful comments. All of them have been taken into consideration when revising the manuscript.
Multiple spelling/grammatical errors exist (e.g., Line 61: "The are also identified…"; Line 321: "There is also nor…").
Response: I have corrected these errors. For example, “They are also identified in plasma after the rats took orally turbid liquor from M. officinalis”; “There is also no recommendations regarding their effective or safe doses for the prophylaxis and treatment of CVDs.”.
Inconsistent italicization of M. officinalis; some instances are italicized while others are not.
Response: I have corrected it. Now, it is “M. officinalis”.
Formatting errors: Line 221 uses incorrect font ("another papers also indicate that…");
Response: I have corrected it.
Line 187 misspells "4-O-methylhonokiol" as "4-0-methylhonokiol."
Response: I have corrected it. Now, it is “4-O-methylhonokiol”.
Line 322: Beyond extraction methods, factors like growth environment, harvesting, and storage of the herb may influence efficacy, thereby indirectly affecting dosage.
Response: I have added more information about it: “Moreover, there is the difference between the extraction process and the relative amount of the chemical compounds in M. officinalis extracts, therefore it is very difficult to compare the dose-effect relationship to the biological materials, and it is a hot topic in further studies. Beyond extraction methods, other factors like growth environment, harvesting, and storage of the herb may influence efficacy, thereby indirectly affecting dosage.”.
The figure oversimplifies mechanisms. Key factors (e.g., AMPK, NF-κB) and their directional roles in cardioprotection are not clearly labeled, despite being discussed in the text.
Response: I have added more information about role of selected key factors on Figure 1. In addition, I have modified the legend to this figure. Now, it is: “The chemical structure of the major bioactive constituents of M. officinalis bark and their potential cardioprotective mechanisms of action. More details about the role of various key factors in cardioprotective mechanism have been described in the text of manuscript.”.
The segment on "The bioavailability and safety of M. officinalis extract…" lacks explicit linkage to cardioprotection. Authors should clarify its relevance to the main theme.
Response: I have added more information about it in the chapter of Conclusion: “This present work reviews newly-available literature regarding the cardiovascular potential especially two individual compounds (magnolol and honokiol) isolated from M. officinalis bark, but this potential is limited to animal in vivo models and in vitro models. Only a single study has examined the cardiovascular properties of M. officinalis extract in obese mice ). In addition, there is no clinical evidence for the absorption and bioavailability of M. officinalis extracts and their main bioactive compounds in human. Moreover, there were not studies simultaneously comparing the activity of magnolol and honokiol. Generally, various studies have taken mixtures of these two phenolic compounds or total phenolic compounds of M. officinalis as research objects. However, there are significant differences in biological activity for these phenolic compounds. For example, Kuribara et al. [1998] suggest that honokiol has stronger anti-anxiety effects than magnolol. Therefore, there is a need for such studies comparing the cardioprotective action of magnolol and honokiol. There is also no recommendations regarding their effective or safe doses for the prophylaxis and treatment of CVDs.”.
“No concrete clinical experiments have investigated the interaction of M. officinalis preparations, including supplements and their bioactive components with various drugs using in the prophylaxis and treatment of CVDs. In the future, more emphasis should be put on human studies and clinical trials.”.
Reviewer 3 Report
Comments and Suggestions for Authors
Line 15: sentence seems incomplete “while, ….”
Line 53: instead of “daily life” how about using the term “daily food”?
Line 59: I would suggest cardioprotective “substances” instead of “protectors” in terms of wording
Line 60: is that meaning that magnolol and honokiol could serve as biomarkes for the existence of other herbs consumed? Pls clarify
Line 79: is reading just the abstract considered adequate enough. I would advise to study the discussion/conclusion chapters, also
Line 99ff: could this effective dose (to mice) be converted/calculated to suggested doses for humans’ Is it in line with line 272 i.e recommended 3-10g which is contrary with the statement that there are no recommendations available line 322. Are there recommendations or not? Pls clarify. Which doses do contain the available related supplements on the Polish market (line 310)
Line 99 ff: and what about possible side effects to humans since no side effects in rats as noted in line 275, but cytotoxicity is mentioned inf line 289
Line 85: the author could summarize all the review findings (of this chapter) in a table, for the reader to get an overview, since to many information. In addition to fig 1
Line 272 vs line 279: The different quantities state, cause a confusion to the reader, since 3-10 grams is not equivalent to 250mg*3 daily, Pls clarify also in regard to the 1.64mg/kg/day in line 283
Author Response
Response: Thank you for reviewing the manuscript and providing such helpful comments. All of them have been taken into consideration when revising the manuscript.
Line 15: sentence seems incomplete “while, ….”
Response: I have corrected this sentence. Now, it is: “Magnolia bark is the main medicinal part of this plant.”.
Line 53: instead of “daily life” how about using the term “daily food”?
Response: I have corrected this sentence. Now, it is: “It is important that it is often available in human in daily food...”.
Line 59: I would suggest cardioprotective “substances” instead of “protectors” in terms of wording
Response: I have corrected this sentence. Now, it is: “honokiol and magnolol are known as cardioprotective substances…”.
Line 60: is that meaning that magnolol and honokiol could serve as biomarkes for the existence of other herbs consumed? Pls clarify
Response: I have added only short information about it, because Niu et al. (2021) have only described this: “Moreover, these two chemical compounds are often used as the markers to analyze the intracorporeal process of single herbs or compound herbal formulae. They are also identified in plasma after the rats took orally turbid liquor from M. officinalis”.
Line 79: is reading just the abstract considered adequate enough. I would advise to study the discussion/conclusion chapters, also
Response: I have corrected the abstract. Now, it is: “The genus Magnolia has been found to exert different biological properties, including antioxidant, anti-cancer, and other. For example, Magnolia officinalis is a classical traditional herb used in various Asian countries, especially China, South Korea, and Japan. Magnolia bark is the main medicinal part of this plant. This paper reviews the current state of knowledge regarding the M. officinalis bark and its active constituents, especially magnolol and honokiol, with a special emphasis on their cardioprotective activity in various models. This review paper also sheds new light on the cardioprotective mechanism of magnolol and honokiol. However, their cardioprotective potential is limited to animal in vivo models and in vitro models. Only a single study has examined the cardiovascular properties of M. officinalis extract in obese mice. In addition, there is no clinical evidence for the absorption and bioavailability of M. officinalis extracts and their main bioactive compounds in human. Moreover, there were not studies simultaneously comparing the activity of magnolol and honokiol. Generally, various studies have taken mixtures of these two phenolic compounds or total phenolic compounds of M. officinalis as research objects. However, there are significant differences in biological activity for these phenolic compounds. For example, Kuribara et al. [1998] suggest that honokiol has stronger anti-anxiety effects than magnolol. Therefore, there is a need for such studies comparing the cardioprotective action of magnolol and honokiol. There is also no recommendations regarding their effective or safe doses for the prophylaxis and treatment of CVDs.”
Line 99ff: could this effective dose (to mice) be converted/calculated to suggested doses for humans’ Is it in line with line 272 i.e recommended 3-10g which is contrary with the statement that there are no recommendations available line 322. Are there recommendations or not? Pls clarify. Which doses do contain the available related supplements on the Polish market (line 310)
Response:
Line 272. It is information about M. officinalis extract: “According to the Chinese Pharmacopoeia (version 2020), the recommended dosage of M. officinalis is 3-10 g [Commision, 2023].”
Line 322. It is information about honokiol and magnolol: “For example, Kuribara et al. [1998] suggest that honokiol has stronger anti-anxiety effects than magnolol. Therefore, there is a need for such studies comparing the cardioprotective action of magnolol and honokiol. There is also no recommendations regarding their effective or safe doses for the prophylaxis and treatment of CVDs.”
Line 310. I have added more information about dietary supplements: “Recently, Siudem et al. [2025] studied the quality and quantity of seven dietary supplements (E1 – E7) containing magnolia bark and its preparations, especially those available in Polish market. These supplements had various forms: tablets, capsules, powder, and bark). Authors identified the active substances: magnolol and honokiol in all tested preparations using high-performance liquid chromatography (HPLC), and nuclear magnetic resonance spectroscopy (1H NMR). In addition, their results indicate that both techniques provide similar results and can be used for quality control of magnolia dietary supplements. For example, tested supplement (E2, as capsules) contained 225 mg of M. officinalis bark extract (202.5 mg honokiol). Other supplement (E3, as capsules) contained 400 mg M. officinalis. Tested supplement (E5) as powder had 40% honokiol and 50% magnolol [Siudem et al., 2025].
Line 99 ff: and what about possible side effects to humans since no side effects in rats as noted in line 275, but cytotoxicity is mentioned inf line 289
Response:
Line 275. It is information about M. officinalis extract: “For example, Liu et al. [2007] noted there is no-adverse effects of M. officinalis for dosage above 240 mg/kg/day (for 90 days) in rats.”.
Line 289. It is information about magnolol: “It is an important that various studies have demonstrated that magnolol has cytotoxicity [Karki et al., 2013A and B; Wei et al., 2014; Wu et al., 2014; Zhang et al., 2017; Hsieh et al., 2018; Chen et al., 2019]. For example, magnolol (40 µM) possessed cytotoxicity on vascular smooth muscle cells [Karki et al., 2013A and B].”.
Line 85: the author could summarize all the review findings (of this chapter) in a table, for the reader to get an overview, since to many information. In addition to fig 1
Response: Now, Table 1 is in the end of chapter – “Cardiovascular properties of M. officinalis extract and its bioactive compounds”. Moreover, I have added more information about these results: “Cardioprotective action of M. officinalis extracts and their main bioactive compounds: magnolol, honokiol and 4-methoxyhonokiol in various models is described in Table 1. Table 1 demonstrates that cardioprotective potential is limited to animal in vivo models and in vitro models. In addition, only a single study has examined these properties of M. officinalis extract in obese mice.”.
I have also added more information about role of selected key factors on Figure 1. In addition, I have modified the legend to this figure. Now, it is: “The chemical structure of the major bioactive constituents of M. officinalis bark and their potential cardioprotective mechanisms of action. More details about the role of various key factors in cardioprotective mechanism have been described in the text of manuscript.”.
Line 272 vs line 279: The different quantities state, cause a confusion to the reader, since 3-10 grams is not equivalent to 250mg*3 daily, Pls clarify also in regard to the 1.64mg/kg/day in line 283
Line 272 and 279. It is information about M. officinalis extract. However, according to the Chinese Pharmacopoeia (version 2020), the recommended dosage of M. officinalis is 3-10 g [Commision, 2023] (line 272). Other authors used other dosages of M. officinalis (line 279). For example, “Oral administration of capsuled extracts of M. officinalis and Phellodendron amurense (250 mg, three times/day, for 6 weeks) was well tolerated in both healthy and obese patients [Garrison and Chambliss, 2006]. However, oral administration of 5-10 g/kg of M. officinalis extract (for 14 days) decreased renal and liver function in rats, but authors did not describe the phytochemical characteristic of used extract and used solvent [Yang et al., 1998].”.
Line 283. It is information about magnolol: “The safe dose of magnolol available from teenage is up to 1.64 mg/kg per day. However, due to the low water solubility and quick metabolism, it has a low bioavailability [Zhang et al., 2019].”.
Reviewer 4 Report
Comments and Suggestions for Authors
The presented manuscript summarizes studies on the cardioprotective effect of M. officinalis extracts and the main metabolites from the plant. The work is interesting but focuses mainly on the effect of metabolites. The research methodology is correctly presented. The title corresponds to the content.
Comment 1: Honokiol and magnolol are plant metabolites with significant cardioprotective potential. However, review papers have already been published on this topic (for instant: DOI: 0.2174/1389450120666191024175727; Yuan, Y., Zhou, X., Wang, Y., Wang, Y., Teng, X., & Wang, S. (2020). Cardiovascular modulating effects of Magnolol and Honokiol, two polyphenolic compounds from traditional Chinese medicine-Magnolia officinalis. Current Drug Targets, 21(6), 559-572.). These paper do not exclude the possibility of writing a new review manuscript but require the authors of the manuscript to refer to previously published review papers. The authors should justify taking up the topic more precisely, point out more specifically a gap in knowledge and write why their work is better (or why fill the gap -if the gap is) than previously published review works in this field, especially since they indicate in the conclusions that “this review paper sheds new light on the cardioprotective mechanism of magnolol and honokiol” -line 331-332
Comment 2: line 316: The authors indicate that the presented manuscript "presents new data" regarding the cardiovasular potential...". The presented work is not an original work but a review, based on existing studies. So what new data does it present - please justify the use of the phrase "new data".
Comment 3: Obesity is one of the risk factors for cardiovascular diseases. However, the authors should distinguish studies on the purely cardioprotective effect (of the extract or isolated substances) from related studies related to obesity and more complex mechanisms, indirectly influencing cardioprotection.
Comment 4: The authors refer in two places in the manuscript to studies on the composition of plant extracts, in which the ingredient was the extract of Magnolia officinalis. One is the study by Yimam et al., [2017] – lines 91-99; the second: lines 278-280 is a study on the oral administration of M. officinalis and Phellodendron amurense extracts. (Additionally, the second study is uncorrect cited in the manuscript, it should probably be the work: Garrison, R., & Chambliss, W. G. (2006). Effect of a proprietary Magnolia and Phellodendron extract on weight management: a pilot, double-blind, placebo-controlled clinical trial. Alternative therapies in health and medicine, 12(1), 50-55.).
However, studies where several extracts are administered simultaneously do not indicate the activity of any one of them separately. Activity may be demonstrated by each of them individually or it may be the effect of interactions between components (positive or negative) therefore these are not good examples of M. officinalis activity to cite in the presented manuscript.
Moreover, introducing these data [Yimam et al., [2017] – lines 91-99} already at the beginning of the chapter on activity creates a kind of mess and distracts from the main topic of the article.
Comment 5: When the phrase "M. officinalis extract" appears in the paper, it should be specified what kind of extract it is (what solvent was used as the extractant).
Comment 6: The paper focuses mainly on two metabolites, such as honokiol and magnolol - their structures are presented in Figure 1. The paper also contains information about 4-O-methylhonokiol (4-methoxyhonokiol), so it is worth presenting the structure of this compound in the figure.
A figure with the structures of the compounds should be included in the introduction, after identifying the main components of the M. officinalis extract.
Comment 7: The authors created Table 1, but they refer to it only in the conclusion. Maybe it is worth doing it earlier.
Comment 8: The authors should refer to literature data and state in the conclusions whether there were studies (and if so how many) simultaneously comparing the activity of magnolol and honokiol. Is it possible to determine which of these compounds has a higher potential in a specific direction or is there a need for such studies.
Comment 9: The authors should cite the reference immediately after the information appears in the text, and not at the end of the paragraph (for instant lines 30-42).
Comment 10: In addition, the authors should carefully check the correctness of citations in the entire manuscript.
Comment 11: A minor note: There are editorial errors in the text: incorrect capitalization, italics, missing letters (line 222, 262, 542) -
Author Response
The presented manuscript summarizes studies on the cardioprotective effect of M. officinalis extracts and the main metabolites from the plant. The work is interesting but focuses mainly on the effect of metabolites. The research methodology is correctly presented. The title corresponds to the content.
Response: Thank you for reviewing the manuscript and providing such helpful comments. All of them have been taken into consideration when revising the manuscript.
Comment 1: Honokiol and magnolol are plant metabolites with significant cardioprotective potential. However, review papers have already been published on this topic (for instant: DOI: 0.2174/1389450120666191024175727; Yuan, Y., Zhou, X., Wang, Y., Wang, Y., Teng, X., & Wang, S. (2020). Cardiovascular modulating effects of Magnolol and Honokiol, two polyphenolic compounds from traditional Chinese medicine-Magnolia officinalis. Current Drug Targets, 21(6), 559-572.). These paper do not exclude the possibility of writing a new review manuscript but require the authors of the manuscript to refer to previously published review papers. The authors should justify taking up the topic more precisely, point out more specifically a gap in knowledge and write why their work is better (or why fill the gap -if the gap is) than previously published review works in this field, especially since they indicate in the conclusions that “this review paper sheds new light on the cardioprotective mechanism of magnolol and honokiol” -line 331-332
Response: I have added more information about it (chapter of Introduction): “In addition, honokiol and magnolol are known as cardioprotective protectors, but only few review papers have been published on this topic [Tsai et al., 1996; Ho and Hong, 2012; Luo et al., 2019; Yuan et al., 2020]. The aim of the present review is to pro-vide an overview of the cardioprotective potential of extracts from M. officinalis bark and their chemical constituents, not only magnolol and honokiol, but also their derivatives in various models. In addition, cardioprotective mechanisms of M. officinalis and its main chemical constituents were reviewed in this paper to provide a basis for further study and development. It is especially important that cardiovascular diseases (CVDs) are a large class of disease, including hypertension, arrhythmia, heart disease, coronary artery disease, and other, and the incidence of CVDs is gradually increasing.”
I have also modified (chapter of Conclusion): “Despite this, this review paper sheds new light on the cardioprotective mechanism of magnolol, honokiol and their derivatives (for example 4-O-methylhonokiol), but further studies are needed to clarity the mechanisms of their action.”
Comment 2: line 316: The authors indicate that the presented manuscript "presents new data" regarding the cardiovasular potential...". The presented work is not an original work but a review, based on existing studies. So what new data does it present - please justify the use of the phrase "new data".
Response: I have changed this phrase. Now, it is: “This present work reviews newly-available literature regarding the cardiovascular potential especially two individual compounds (magnolol and honokiol) isolated from M. officinalis bark, but this….”.
Comment 3: Obesity is one of the risk factors for cardiovascular diseases. However, the authors should distinguish studies on the purely cardioprotective effect (of the extract or isolated substances) from related studies related to obesity and more complex mechanisms, indirectly influencing cardioprotection.
Response: I have prepared new chapter – “Anti-obesity actions (in vitro and in vivo model)”
Comment 4: The authors refer in two places in the manuscript to studies on the composition of plant extracts, in which the ingredient was the extract of Magnolia officinalis. One is the study by Yimam et al., [2017] – lines 91-99; the second: lines 278-280 is a study on the oral administration of M. officinalis and Phellodendron amurense extracts. (Additionally, the second study is uncorrect cited in the manuscript, it should probably be the work: Garrison, R., & Chambliss, W. G. (2006). Effect of a proprietary Magnolia and Phellodendron extract on weight management: a pilot, double-blind, placebo-controlled clinical trial. Alternative therapies in health and medicine, 12(1), 50-55.).
Response: I have added the paper of Garrison and Chambliss (2006): “Oral administration of capsuled extracts of M. officinalis and Phellodendron amurense (250 mg, three times/day, for 6 weeks) was well tolerated in both healthy and obese patients [Garrison and Chambliss, 2006].”.
Moreover, I have added new review papers: “In addition, various plant preparations, including extracts have often anti-obesity potential [Al-Snafi and Alfuraiji, 2023; Saglam and Sekerler, 2024].
Al-Snafi, A.E., Alfuraiji, N. Medicinal plants with anti-obesity effects: a special emphasis on their mode of action. Bahrain Med Bull 2023; 2: 1-7.
Saglam, K., Sekerler, T. A compherensive review of the anti-obesity properties of medicinal plants. Pharmed 2024; 1: 1-23.
However, studies where several extracts are administered simultaneously do not indicate the activity of any one of them separately. Activity may be demonstrated by each of them individually or it may be the effect of interactions between components (positive or negative) therefore these are not good examples of M. officinalis activity to cite in the presented manuscript.
Response: I have added comments for this experiment: “Yimam et al. [2017] indicate that the extract from Morus alba, Yerba mate and M. officinalis extract has a significant effect on weight loss in high-fat mice. In this experiment, plant extract was administered at oral doses of 300 mg/kg, 450 mg/kg and 600 mg/kg for 7 weeks; orlistat (40 mg/kg/day) was used as a positive control. Statistically significant changes in body weight (decreased by 9.1, 19.6 and 25.6% compared to the control group at week-7) were observed for mice treated with used extract at 300, 450 and 600 mg/kg, respectively. Reductions in total cholesterol, in triglyceride, and in low-density lipoprotein (LDL) were observed for plant extract at all three doses (300, 450 and 600 mg/kg). In this experiment, M. officinalis stem bark was extracted by a supercritical fluid and further crystalized to give a mixture of magnolol and honokiol with content higher than 95%. However, this study where three extracts are administered simultaneously do not indicate the activity of any one of them separately. Activity may be demonstrated by each of them individually or it may be the effect of interactions between components (positive or negative) therefore it is difficult to indicate on anti-obesity activity of M. officinalis extract.”.
Moreover, introducing these data [Yimam et al., [2017] – lines 91-99} already at the beginning of the chapter on activity creates a kind of mess and distracts from the main topic of the article.
Response: I have added comments for this experiment: “Yimam et al. [2017] indicate that the extract from Morus alba, Yerba mate and M. officinalis extract has a significant effect on weight loss in high-fat mice. In this experiment, plant extract was administered at oral doses of 300 mg/kg, 450 mg/kg and 600 mg/kg for 7 weeks; orlistat (40 mg/kg/day) was used as a positive control. Statistically significant changes in body weight (decreased by 9.1, 19.6 and 25.6% compared to the control group at week-7) were observed for mice treated with used extract at 300, 450 and 600 mg/kg, respectively. Reductions in total cholesterol, in triglyceride, and in low-density lipoprotein (LDL) were observed for plant extract at all three doses (300, 450 and 600 mg/kg). In this experiment, M. officinalis stem bark was extracted by a supercritical fluid and further crystalized to give a mixture of magnolol and honokiol with content higher than 95%. However, this study where three extracts are administered simultaneously do not indicate the activity of any one of them separately. Activity may be demonstrated by each of them individually or it may be the effect of interactions between components (positive or negative) therefore it is difficult to indicate on anti-obesity activity of M. officinalis extract.”.
Comment 5: When the phrase "M. officinalis extract" appears in the paper, it should be specified what kind of extract it is (what solvent was used as the extractant).
Response: I have added more information about solvent, but authors often do not described the phytochemical characteristic of used plant extract and used solvents. For example, I have added: “In this experiment, M. officinalis extract was prepared by Bioland Co., Ltd (Korea) and dissolved in 0.5% ethanol (Sun et al., 2014).; ”In this experiment, M. officinalis stem bark was extracted by a supercritical fluid and further crystalized to give a mixture of magnolol and honokiol with content higher than 95% (Yimam et al., 2017).
Comment 6: The paper focuses mainly on two metabolites, such as honokiol and magnolol - their structures are presented in Figure 1. The paper also contains information about 4-O-methylhonokiol (4-methoxyhonokiol), so it is worth presenting the structure of this compound in the figure.
Response: I have added the chemical structure of 4-O-methylhonokol on Figure 1.
A figure with the structures of the compounds should be included in the introduction, after identifying the main components of the M. officinalis extract.
Response: I have added Figure 1 in the introduction. Fig. 1 - The chemical structure of the major bioactive constituents of M. officinalis bark and their potential cardioprotective mechanisms of action
Comment 7: The authors created Table 1, but they refer to it only in the conclusion. Maybe it is worth doing it earlier.
Response: Now, Table 1 is in the end of chapter – “Cardiovascular properties of M. officinalis extract and its bioactive compounds”.
Comment 8: The authors should refer to literature data and state in the conclusions whether there were studies (and if so how many) simultaneously comparing the activity of magnolol and honokiol. Is it possible to determine which of these compounds has a higher potential in a specific direction or is there a need for such studies.
Response: I have added this information in the chapter of conclusion: “In addition, there is no clinical evidence for the absorption and bioavailability of M. officinalis extracts and their main bioactive compounds in human. Moreover, there were not studies simultaneously comparing the activity of magnolol and honokiol. Generally, various studies have taken mixtures of these two phenolic compounds or total phenolic compounds of M. officinalis as research objects. However, there are significant differences in biological activity for these phenolic compounds. For example, Kuribara et al. [1998] suggest that honokiol has stronger anti-anxiety effects than magnolol. Therefore, there is a need for such studies comparing the cardioprotective action of magnolol and honokiol. There is also no recommendations regarding their effective or safe doses for the prophylaxis and treatment of CVDs.”.
Comment 9: The authors should cite the reference immediately after the information appears in the text, and not at the end of the paragraph (for instant lines 30-42).
Response: I have corrected it.
Comment 10: In addition, the authors should carefully check the correctness of citations in the entire manuscript.
Response: I have corrected it.
Comment 11: A minor note: There are editorial errors in the text: incorrect capitalization, italics, missing letters (line 222, 262, 542) –
Response: I have corrected these errors.
Round 2
Reviewer 4 Report
Comments and Suggestions for Authors
The changes provided to the text have improved the quality of the manuscript. In my opinion, the manuscript can be published in its current form.